# XPLAINLLM: A QA EXPLANATION DATASET FOR UNDERSTANDING LLM DECISION-MAKING

## ABSTRACT

Large Language Models (LLMs) have recently made impressive strides in natural language understanding tasks. Despite their remarkable performance, understanding their decision-making process remains a big challenge. In this paper, we look into bringing some transparency to this process by introducing a new explanation dataset for question answering (QA) tasks that integrates knowledge graphs (KGs) in a novel way. Our dataset includes 12,102 question-answer-explanation (QAE) triples. Each explanation in the dataset links the LLM's reasoning to entities and relations in the KGs. The explanation component includes a *why-choose* explanation, a *why-not-choose* explanation, and a set of *reason-elements* that underlie the LLM's decision. We leverage KGs and graph attention networks (GAT) to find the *reason-elements* and transform them into *why-choose* and *why-not-choose* explanations that are comprehensible to humans. Through quantitative and qualitative evaluations, we demonstrate the potential of our dataset to improve the in-context learning of LLMs, and enhance their interpretability and explainability. Our work contributes to the field of explainable AI by enabling a deeper understanding of the LLMs decision-making process to make them more transparent and thereby, potentially more reliable, to researchers and practitioners alike. Our dataset is available at: `http://anonymous.4open.science/r/XplainLLM`

## 1 INTRODUCTION

Large Language Models (LLMs) (Kenton & Toutanova, 2019; Liu et al., 2019; Brown et al., 2020; Anil et al., 2023) have significantly influenced Natural Language Understanding (NLU) (Liu et al., 2021; 2023), leading to performance improvements in various tasks. As these models continue to make progress, it is important to understand the rationale behind their decision-making (Arrieta et al., 2020). A deeper comprehension of the LLM decision-making process is crucial to fostering trust in their predictions, enabling the design of more robust and reliable AI systems for end users.

Development of explainable AI (XAI) as an area of research has seen the emergence of a number of methods that seek to explain the decision-making processes of machine learning models. Such methods span from local explanation techniques such as LIME (Ribeiro et al., 2016) and SHAP (Lundberg & Lee, 2017), to global explanation strategies such as feature importance (Casalicchio et al., 2019). Despite substantial progress in explaining machine learning models, these methods under-perform when deployed for LLMs, particularly in complex tasks such as question-answering (QA), making the output difficult for humans to understand. The inherent complication and lack of transparency in LLMs (Wu et al., 2022), combined with context-rich commonsense reasoning, necessitates constructing more human-understandable and comprehensive explanations to faithfully interpret their predictions.

Current explanation methods for LLMs primarily focus on attention mechanisms (Clark et al., 2019; Bills et al., 2023) and feature-based interpretations (Jacovi et al., 2021). The former approach bases the explanations on self-attention weights in models like BERT (Kenton & Toutanova, 2019) and GPT-2 (Radford et al., 2019), and deduces correlations between input tokens and the model's predictions. However, the relationships highlighted in these generated explanations are difficult to understand for humans. Moreover, attention can be difficult to interpret due to the typically complex inter-layer interactions, and may not align the relative importance of tokens in the model's reasoning process (Hahn, 2020; Sajjad et al., 2022). Feature-based explanation methods, in contrast, aim to

quantify the contribution of individual features or tokens to the model output, but they fail to capture the wider context and relations that are fundamental to understanding the model's reasoning (Molnar et al., 2022).

We introduce XplainLLM, a dataset that addresses the growing need for transparency, explainability, and understandability in the decision-making processes of LLMs. By integrating knowledge graphs (KGs) and graph attention networks (GAT) (Veličković et al., 2018), our dataset provides a human-understandable explanation of LLM decision-making in QA tasks. We link the LLM reasoning process to the entities and relations within KGs to help provide an intuitive and interpretable representation of the LLM's decision-making process. Through the linked connections, researchers can gain deeper insights into the underlying rationale for predictions. Our process can also help facilitate model tuning, debugging, robustness evaluation and demonstration in in-context learning. XplainLLM includes 12,102 question-answer-explanation (QAE) triples and aims to drive improvements in both the performance and explainability of LLMs. Our evaluation shows that LLMs with explanations enhance performance by an average of 2.4% when decision-making knowledge is transferred between LLMs. LLMs with explanations outperform the benchmark, with a performance gap extending to 17%. The overall quality of explanations achieves an average score of 0.87/1 by human evaluators, and an average of 0.89/1 by automated evaluators.

**Contributions and significance of the dataset**    In this paper, we make several key contributions to the field of explainable AI for LLMs:

- Bridging the Gap with Natural Language: To the best of our knowledge, XplainLLM is the first dataset to capture the most influential elements behind the model reasoning and present the decision-making process through human-understandable explanations. Our dataset extends beyond merely explaining the "why"; we emphasize the "why not", introducing a novel paradigm in transparent AI explainability.

- Comprehensive Understanding of Model Reasoning: XplainLLM incorporates *reason-elements* from KGs, top-ranked *reason-elements*, *why-choose* and *why-not-choose* explanations. The goal is to empower the community to delve deeper into the decision-making dynamics of LLMs. This work contributes to enhancing the knowledge and the transparency of LLM reasoning.

- Aligning Human Understanding and Model Explainability: XplainLLM organizes the decision-influencing elements into coherent natural language sentences. Our explanations can be used in reinforcement learning from human feedback (RLHF) (Christiano et al., 2017), to support related research. We evaluate the quality of our explanations through both automated and human evaluations, and the results underscore our dataset's quality on multiple metrics.

## 2    RELATED WORK

**Reasoning in LLMs**    XAI aims to address the issue of interpreting the outcomes of language models (Li et al., 2023; Wiegreffe et al., 2021; Madsen et al., 2022). One of its goals is to generate explanations that enable humans to easily understand the decision-making process. Zelikman et al. (2022) introduce a method that iteratively generates the rationales step-by-step. Huang et al. utilize the chain-of-thought (CoT) to find the rationale and apply the reasoning capabilities of LLMs to robotic planning tasks. However, these explanations are inherently constrained in capturing prompt-specific reasoning, limiting generalization to out-of-distribution scenarios and potentially missing the decision-making process that our work focuses on.

Another goal is focused on explaining in a trustworthy way. Rajani et al. (2019a) introduces an explainable factor to minimize the risk of unreasonable explanation generation. Chen et al. (2021) integrate the external knowledge to generate why and why-not counterfactual explanations. Zelikman et al. (2022) apply self-checker mechanism to ensure trusted rationals. However, these methods, while enhancing performance or providing external explanations, fail to accurately capture the core reasoning of LLMs. In contrast, our work enhances LLM trustworthiness and deepens human understanding of its decision-making, improving the potential in end-user applications.

| Dataset | Size | Answer Format | Expl. Format | Source | Model Match? | Self-Explanatory? | "Why Not" Included? |
|---------|------|---------------|--------------|--------|:------------:|:-----------------:|:-------------------:|
| CoS-E | 9,500 | MC | NL | Human | × | × | × |
| ECQA | 10,962 | MC | NL | Human | × | ✓ | × |
| Neuron | 307,200 | Neuron | NL + Score | Model | ✓ | × | × |
| XplainLLM | 12,102 | MC | NL | Model | ✓ | ✓ | ✓ |

Table 1: Comparison of prevalent explanation datasets with ours, detailing instance count (Size), answer types (Answer Format: e.g., multiple-choice (MC)), explanation styles (Explanation Format: e.g., natural language (NL)), origin (Source), alignment with model reasoning (Model Match?), necessity of human intervention to deduce the reasoning (Self-Explanatory?), and inclusion of reasons for alternative answer rejection ("Why Not" Included?).

**Explanation Datasets** The explainable datasets for language models can be categorized into three types (Wiegreffe & Marasovic, 2021): (1) highlights: provide input elements such as words and phrases, as explanations to a predicted output (Camburu et al., 2018; DeYoung et al., 2020; Yin et al., 2021; Bills et al., 2023); (2) free-text explanations: provide readable textual explanations in words or sentences (Rajani et al., 2019b; Sap et al., 2020; Brahman et al., 2021); (3) structured explanations: provide natural language explanation but are constrained by the explanation writing process (Aggarwal et al., 2021; Jhamtani & Clark, 2020; Inoue et al., 2020). Different from these, our explanation incorporates highlighted elements and guided instruction to generate a free-text explanation. This unique combination can enhance both the trustworthiness and comprehensiveness of the content. We present a comparison with prevalent explanation datasets ((Rajani et al., 2019b; Aggarwal et al., 2021; Bills et al., 2023)) in Table 1.

## 3 METHODOLOGY

We choose QA as the context for studying the decision-making process of LLMs, as questions facilitate an intuitive understanding of tasks and models. Given a pre-trained LLM $\mathbb{M}$, our input content $Z$ includes question $Q$ and a set of $i$ possible answer choices $A = \{a_1, a_2, .., a_i\}$. We denote the answer chosen by the $\mathbb{M}$ with $y$. Our goal is to find an explanation $\mathbb{E}_{why}$ for why $\mathbb{M}$ chooses a certain answer, and an explanation $\mathbb{E}_{whynot}$ for why $\mathbb{M}$ does not choose the other options.

We introduce a GAT-based method to explain the decision-making process of $\mathbb{M}$. We first tokenize the combined sequence of $Q$ and $A$ into a content elements set $X = \{x_1, x_2, ..., x_j\}$, where $j$ is the number of elements. Consider a graph $g_e$ with layers $L = \{l_1, l_2, ..., l_k\}$, and nodes $E = \{e_1, e_2, ..., e_n\}$, where $k$ denotes the number of layers and $n$ represents the node count. The nodes and edges are constructed by pruning the retrieved sub-graph $g_k$ from the KG, guided by the input content. We integrate $g_e$ and $\mathbb{M}$ for the final decision. Through a GAT model $\mathbb{G}$, we obtain the decision representations and convert them into meaningful, human-understandable explanations. We delve into each step in the subsequent sections.

### 3.1 EXPLAINER MODEL

Our explainer model makes novel use of retrieved KG and GAT. Its architecture is illustrated in Figure 1, and consists of three major stages: (a) graph construction, (b) decision interpretation, and (c) controlled explanation generation. In this paper, interpretation refers to understanding of the model's weights by humans, e.g., attentions of concrete nodes or elements, or the weights in the model. Explanation refers to explaining the model's decision-making process in a manner comprehensible to humans.

**Graph Construction.** To capture the pivotal components that impact the reasoning of $\mathbb{M}$, we construct a multi-relational graph, which fuses knowledge of $\mathbb{M}$ and $g_k$. Given the input $Q$ and $A$, we follow (Yasunaga et al., 2021) to construct this graph, yielding an element-graph $g_e \subseteq g_k$. We begin with question entities $E_Q$ and answer entities $E_A$, both subsets of $g_k$, indicating their respective nodes can be located within $g_k$. The $g_k$ is initially extracted from a KG, guided by $E_A$ and $E_Q$, and includes $k$-hop neighbors. Subsequently, $g_k$ is pruned based on the *relevance score* to yield $g_e$. Algorithm 1 (Appendix A.1) presents this procedure. $g_e$ serves as a compact and informative representation of the important elements and relations in the decision-making process.

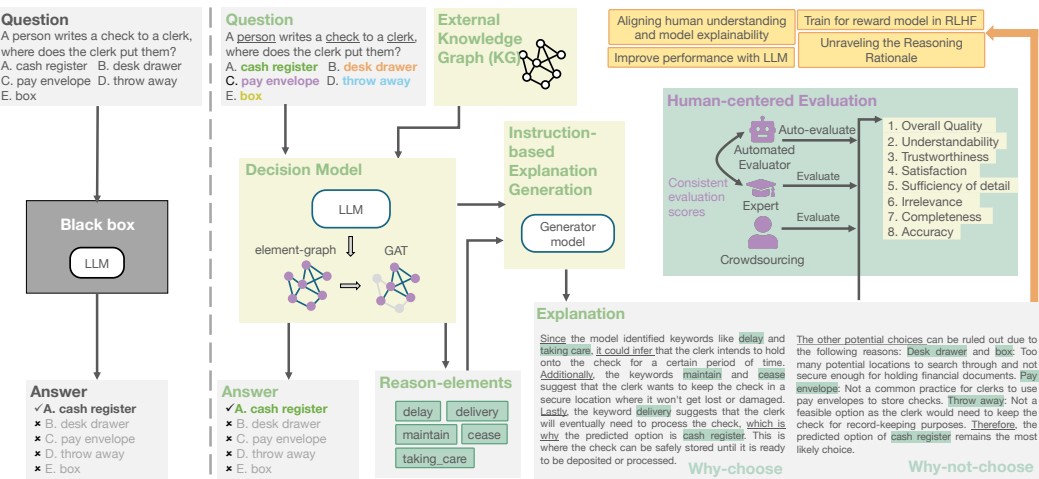

Figure 1: The data collection, processing and evaluation of XplainLLM dataset. In contrast to traditional black-box LLMs, our approach leverages *external KG* and *GAT* to interpret the decision-making process of LLMs and extract the *reason-elements*. A generator model is involved in instruction-based generation for *why-choose* and *why-not-choose* explanations. We *evaluate* the explanations with experts, crowdsourcing and automated evaluators in eight various dimensions. The explanations can benefit XAI, LLMs, RLHF and model understanding.

**Decision Interpretation.** Given the $g_e$, we use GAT model $\mathbb{G}$ to obtain the representation of the decision-making process. Consider any node $e_t$ in $g_e$, its neighboring nodes are denoted by $\mathcal{N}(e_t)$, with a specific neighbor as $e_s \in \mathcal{N}(e_t)$. Each node has a feature-embedding $\boldsymbol{h}_t^l$ at layer $l$, representing its semantic features.

The feature-embedding $\boldsymbol{h}_e^l$ of $e$ is computed by the relevance score and three kinds of node embeddings: (1) node type-embeddings $u_t$; (2) node feature-embeddings $\boldsymbol{h}_t^{l-1}$; and (3) relation-embeddings $r_t$. The $\boldsymbol{h}_t^l$ is calculated as following:

$$\boldsymbol{h}_t^l = f_e\Big( \sum_{e_s \in \mathcal{N}(e_t) \cup \{e_t\}} \alpha_{ts} m_{ts} \Big) + \boldsymbol{h}_t^{l-1}, \tag{1}$$

where $\alpha_{ts}$ is the attention mechanism, $m_{ts}$ is the message passing from $e_s$. The $\alpha_{ts}$ is used to discern the critical connections in the decision-making process during the aggregation of message passing, following the approach in (Veličković et al., 2018). The $\alpha_{ts}$ is calculated as following:

$$\alpha_{ts} = softmax(e_t) = \frac{exp(e_t)}{\sum_{v \in \mathcal{N}(e_t)} exp(e_{tv})}. \tag{2}$$

The updated node features at the final layer of the $\mathbb{G}$ are considered as the final representations for major reasoning. Further details of $e_t$, $u_t$, $r_t$ and $m_{ts}$ can be found in Appendix B.2.

In the learning and inference stage, the probability of selecting an answer is defined as $P(a|q) \propto exp(MLP(\mathbb{H}^M, \mathbb{H}^{itp}))$, where $\mathbb{H}^{itp} = h_t^l || \Lambda$. Here, $\Lambda$ represents the attention-based pooling of $G_e$, and $\mathbb{H}^M$ denotes the embedding from $\mathbb{M}$. The representation of node $E$ serves as a depiction of the decision-making process. We identify $E$ as the *reason-elements* for the later explanation generation.

**Controlled Explanation Generation.** After obtaining the *reason-elements*, they are converted to human-understandable explanations by (1) extracting key *reason-elements* and (2) generating explanations through instructions.

(1) We use the input content $Z$, predicted answer $y$ of $\mathbb{M}$, the *reason-elements* $E$, and attentions $\alpha$ to identify explanation-critical nodes. The top $m$ nodes, ranked by $\alpha$, are selected and identified as key *reason-elements* set $R$.

(2) We use $R$ to guide the generation of precise and human-understandable explanations. We introduce the generator model $\mathbb{F}$, which imposes a structured format on the explanations: (1) a *why-choose* part

|  | Why-choose | Why-not-choose | Whole Explanation |
|---|---|---|---|
| Overall | 59.92 | 59.48 | 119.39 |
| Training Set | 59.79 | 59.41 | 119.20 |
| Dev Set | 59.47 | 58.51 | 117.98 |
| Testing Set | 61.36 | 60.97 | 122.32 |

Table 2: The average word counts of *why-choose* explanation, *why-not-choose* explanation and whole explanation in our XplainLLM dataset.

and (2) a *why-not-choose* part. The explanations provide the reasoning of $\mathbb{M}$, detailing why specific choices were made and others dismissed. The structures are defined as follows:

$$\textbf{(1) } W : [Z],\ P : [y],\ T : [O],\ C : [\{R|R \in E\}];\ \textbf{(2) } I : [\mathbb{E}_{why}],\ \hat{T} : [O],$$

where $W, P, T, O, C, I$ and $\hat{T}$ are the predefined structure guiding the generation process. Additional details are elaborated in Appendix E.5.3.

## 3.2 DATA PREPARATION

The selection of $\mathbb{M}$ and the dataset plays an important role in studying the decision-making process of LLM. Ideally, we hope our dataset to mirror common daily usage, helping the XAI community in fostering future trust between humans and AI. As the first dataset explaining LLM decision-making process in a human-understandable way, we commence our study from a foundational LLM.

The input question and answer choices are from the CommonsenseQA dataset (Talmor et al., 2019). CommonsenseQA is a dataset about commonsense questions, sourced from human queries. We use RoBERTa-Large (Liu et al., 2019) as our $\mathbb{M}$, fine-tuning it on the official training set of CommonsenseQA. Given its foundational role in the LLM family (Zhou et al., 2023), understanding its reasoning process is valuable. We utilize ConceptNet (Speer et al., 2017) as our KG to obtain $g_k$. This KG captures commonsense concepts and their interrelations. Our $g_e$ is structured as a 5-layer GNN model, and for $\mathbb{F}$, we leverage GPT-3.5-turbo (Ouyang et al., 2022) to provide a natural language explanation in a sentence or a paragraph. To ensure the quality of our dataset, we conduct a post-generation evaluation. All explanations undergo human review. Human evaluators identify inaccuracies, and any discrepancies in explanations, and return to $\mathbb{F}$ for refinement. This procedure mitigates potential issues from model-generated explanations, guaranteeing clarity and relevance aligned with human understanding. Further experimental specifics and data collection procedures are provided in the Appendix E.5.1 and E.5.2.

## 3.3 DATASET DESCRIPTION

**Schema.** XplainLLM contains fields that correspond to the QA pair, the model's predicted answer, the ground-truth label, and an explanation set.

**Explanations Set.** The explanation set includes a set of 50 *reason-elements* $E$, e.g., words or phrases, sorted by attentions, a set of top-5 *reason-elements* $R$, a *why-choose* explanation $\mathbb{E}_{why}$ in free-text form, and a *why-not-choose* explanation $\mathbb{E}_{whynot}$ also in free-text form. An example instance is shown in Appendix D.4.

**Statistics.** XplainLLM includes 12,102 instances of explanations, split according to the official CommonsenseQA's partitioning into three sets: the training, development (dev), and testing sets. The average word count of $\mathbb{E}_{why}$ and $\mathbb{E}_{whynot}$ are 59.92 and 59.48 respectively, resulting in an aggregate count of approximately 119.39 words per whole explanation. A more detailed breakdown of the average word count is provided in Table 2. Additional statistics can be found in Appendix C.3.

**Limitations.** Committed to transparency and rigorous analysis, we acknowledge potential limitations in our dataset. Since our $R$ is originally derived from $g_k$, any inherent limitations or inaccuracies within $g_k$ could influence the quality of our explanations. Additionally, using different $\mathbb{F}$ might yield variations.

**Categorization.** To evaluate the limitations and enhance comprehension of XplainLLM for future applications, we classify the data into four categories according to the explanation set: (1) $R$ is effective for explanation generation; (2) additional knowledge is used in explanation generation; (3)

|          | #Examples | Cat. (1) | Cat. (2) | Cat. (3) | Cat. (4) |
|----------|-----------|----------|----------|----------|----------|
| Overall  | 100       | 69.0%    | 12.0%    | 31.0%    | 4.0%     |
| CP       | 74        | 71.6%    | 10.8%    | 27.0%    | 1.3%     |
| IP       | 26        | 61.5%    | 15.4%    | 42.4%    | 11.5%    |

Table 3: The percentage distribution of the four categories: (1) utilizing *reason-elements*, (2) utilizing alternative knowledge, (3) recognizing irrelevant *reason-elements* and (4) identifying incorrect predicted answer, among 100 examples randomly sampled from our XplainLLM dataset.

$R$ is irrelevant in explanation generation; and (4) $\mathbb{E}_{why}$ or $\mathbb{E}_{whynot}$ indicates an incorrect $y$. Table 3 presents the percentage distribution of the 4 categories among 100 examples randomly sampled from the testing set of XplainLLM, along with statistics for correct predicted answers (CP) and incorrect predicted answers (IP). Note that each example may fall into multiple categories. The percentage of Cat.(3) is lower for CP examples compared to IP examples, indicating the contribution of $R$ to the prediction performance. Additionally, Cat.(4) shows a similar distribution of real distribution, demonstrating the effectiveness of $R$ in improving model performance.

## 4    EXPERIMENTS AND EVALUATION

### 4.1    TASK SETTINGS AND MODELS

We demonstrate the XplainLLM dataset on two tasks: **Task 1** (human-centered evaluation) - evaluating explanation quality, and **Task 2** (objective evaluation) - transferring explanation knowledge in LLMs. Our **baseline task** is QA with no explanation.

**Task 1: Evaluating Explanation Quality. Input =** $(Q, A, y, \mathbb{E}_{why}, \mathbb{E}_{whynot})$**, Output =** $\mathcal{X}$**.**

In this task, we engaged three human experts to evaluate the quality of explanations. Each expert has a graduate-level education (taught in English) and at least three years of research experience in natural language processing (NLP). Additionally, we conducted evaluations with 50 general users through crowdsourcing platforms to gauge their perception of the explanations and understand the potential significance of the explanations in future human-centric applications. Specifically, we utilized the Prolific platform (https://www.prolific.com) for these evaluations. Our participant pool was gender-balanced, comprised of native English speakers with at least a high school education. To mitigate human bias in evaluations, we adhered to the methodology outlined by (Hoffman et al., 2018). We provided detailed instructions and examples to participants, to ensure consistent rating standards. Details of these materials can be found in Appendix F.6. Inspired by EVAL[1], we introduce automated evaluator model $\mathbb{A}$ as complement to this task. We select two automated evaluators, GPT-3.5-turbo and GPT-4, given their superior human-like comprehension and linguistic capabilities. Their evaluation settings are the same as humans, provided with $Q$, $A$, $y$, $\mathbb{E}_{why}$, $\mathbb{E}_{whynot}$, along with instructions and examples. The output $\mathcal{X}$ is a set of normalized scores, standardized to maintain consistent rating standards across various metrics. For a specific score $\mathcal{X}_D$ based on evaluation metric $D$ is computed as follows:

$$\mathcal{X}_D = \mathcal{F}_g(\tilde{s}_D) \tag{3}$$

$$\mathcal{F}_g(\tilde{s}_D | D, \beta) = \left\{ \begin{array}{ll} \tilde{s}_D / \max(s_D), & \beta = 1 \\ \min(s_D) / \tilde{s}_D, & \beta = 0 \end{array} \right. \tag{4}$$

where $\tilde{s}_D$ is an original score given by $\mathbb{A}$ or humans, $\mathcal{F}_g$ is a normalization function, $s_D$ is the rating scale, and $\beta$ is the metric type. Specifically, $\beta = 1$ implies a higher score indicates better performance, whereas $\beta = 0$ suggests a lower score is preferable. In our study, the rating scale bounds are given by $\max(s_D) = 3$ and $\min(s_D) = 1$.

**Task 2: Transferring Explanation Knowledge in LLMs. Input =** $(Q, A, \mathbb{E}_{why})$**, Output =** $\tilde{y}$**.**

In this task, we investigate the potential of transferring explanation knowledge to improve the performance of other LLMs, denoted as $\mathbb{L}$. Given the $Q$, $A$ and $\mathbb{E}_{why}$, $\mathbb{L}$ is to identify the correct answer choice $\tilde{y} \in A$. The task seeks to find

$$\tilde{y} = argmax_{\tilde{y} \in A} \mathcal{P}(\tilde{y} \mid Q, A, \mathbb{E}_{why}) \tag{5}$$

---

[1]Evals is a framework introduced by OpenAI, designed for the automated evaluation of LLMs: https://github.com/openai/evals.

where $\mathcal{P}$ is the probability of answer choice. The goal is to evaluate if the inclusion of decision-making explanations can enhance the accuracy of $\mathbb{L}$'s answers. While models like GPT-3.5 are powerful in many tasks, they may fail in commonsense questions compared to the supervised models (Khashabi et al., 2020; Yasunaga et al., 2021; Zhang et al., 2022b). We evaluate under three settings: vanilla, CoT, and self-consistency (Wang et al., 2022). The vanilla approach processes QA pairs directly. In contrast, CoT leverages examples with human-labeled reasoning steps, and the self-consistency approach employs a majority voting strategy. For this task, we utilize GPT-3.5-turbo as our $\mathbb{L}$.

**Baseline Task: Question Answering without Explanation. Input = $(Q, A)$, Output = $\tilde{y}$.**

In this task, we explore the capabilities of LLMs in a pure QA task. The input only consists of $Q$ and $A$. The LLM $\mathbb{L}$ then produces a predicted answer $\tilde{y}$. The problem is formulated as follows:

$$\tilde{y} = argmax_{\tilde{y} \in A} \mathcal{P}(\tilde{y} \mid Q, A) \tag{6}$$

We use GPT-3.5, BLOOM 176B (Scao et al., 2022), GPT-NeoX (Black et al., 2022), Bloomberg GPT (Wu et al., 2023), and OPT 66B (Zhang et al., 2022a) as $\mathbb{L}$ in this task.

## 4.2 EVALUATION METRICS

We evaluate the quality of explanations from both human and objective perspectives. Our goal is a comprehensive and fair evaluation. We aim to present a thorough and fair evaluation of the explanations. These dual perspectives help us discern the strengths and weaknesses of the explanations, guiding potential directions for their improvement.

**Human-centered Metrics. (Task 1)** We follow (Hoffman et al., 2018) to measure the quality and relevance of the explanations. Each explanation is presented with eight evaluative questions, representing distinct evaluation dimensions. Our human-centered evaluation metrics encompass: overall quality, understandability, trustworthiness, satisfaction, detail sufficiency, irrelevance, completeness, and accuracy. Detailed definitions and the specific survey questions are provided in the Appendix F.6.2 and F.6.1. Evaluators allocate scores to these questions using a three-point Likert scale: 1 (disagree), 2 (neutral), and 3 (agree). Subsequently, scores are normalized by $\mathcal{F}_g$ to the range [0, 1]. Higher scores suggest better quality.

**Objective Metrics. (Task 2, Baseline Task)** We use model accuracy, a foundational metric of performance, to evaluate the ability of LLMs $\mathbb{M}$ or $\mathbb{L}$ in selecting the correct answers. Accuracy is determined by the ratio of questions where the model's selected answer aligns with the ground truth. A higher accuracy implies that the model is more capable of choosing the correct answer among alternatives.

## 5 RESULTS AND DISCUSSION

### 5.1 HUMAN-CENTERED EVALUATION

We conducted human-centered evaluations to go beyond the technical evaluation of the explanations and see how they are rated by humans, to determine potential for use in future human-centered applications. In this evaluation, participants were asked to evaluate a set of 20 QAE triples, randomly selected from our dataset.

### 5.1.1 EXPERT AND AUTOMATED EVALUATION

In this evaluation, the feedback from human experts highlighted the distinctiveness of our explanations compared to previous methods. One expert remarked, "In comparison to prior explanations, these explanations provide a more intuitive understanding of the model's decision-making process. The explanations are cogent, and even in instances of erroneous predictions, the underlying reasoning remains transparent and comprehensible." This feedback underscores the clarity and transparency of our explanations.

The detailed scores provided by human experts are shown in Table 4. Across eight evaluation metrics, the explanations received an impressive average score of 0.93. Notably, the metrics of "understandability" and "completeness" garnered the highest average scores, reflecting the success of our approach in delivering human-understandable insights into the LLM's decision-making process.

|          | Overall Quality | Understandability | Trustworthiness | Satisfaction | Sufficiency of detail | Irrelevance | Completeness | Accuracy |
|----------|-----------------|-------------------|-----------------|--------------|------------------------|-------------|--------------|----------|
| GPT-3.5  | 0.98            | 0.98              | 0.98            | 0.98         | 0.98                   | 0.53        | 0.98         | 0.98     |
| GPT-4    | 0.90            | 0.93              | 0.87            | 0.87         | 0.88                   | 0.69        | 0.87         | 0.88     |
| Human Expert | 0.91        | 0.97              | 0.95            | 0.89         | 0.98                   | 0.85        | 0.97         | 0.93     |

Table 4: Evaluation by automated evaluator GPT-3.5, GPT-4, and human experts, on 8 evaluation metrics.

However, it's worth noting that the metric of "irrelevance" received a slightly lower score of 0.85, suggesting that there might be instances where our explanations include some irrelevant details. This is an area we aim to refine in the future work.

The automated evaluator $\mathbb{A}$ follows in-context learning to simulate human expert evaluation. Both GPT-3.5 and GPT-4 demonstrated a commendable ability to discern the quality of explanations. The results show in Table 4 and Figure 2. Notably, the performance of these automated evaluators aligns closely with human expert evaluations across most dimensions.

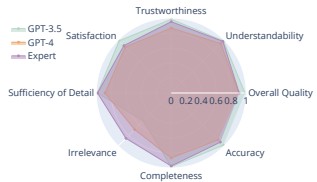

Further insights into the human-like understanding of automated evaluators and their assessment of explanations are detailed in Table 5. This data shows a significant agreement between the automated evaluators and human experts. The scores assigned by $\mathbb{A}$ correlate with those given by human experts, underscoring $\mathbb{A}$'s adeptness in evaluating explanations. Such findings further support the credibility and value of our explanations.

Figure 2: Evaluation by human experts, automated evaluator GPT-3.5 and GPT-4.

### 5.1.2 Crowdsourcing Evaluation

The average scores from crowdsourcing on eight metrics are shown in Figure 3. We show the average score of the overall explanations, explanations for correct predictions (CP), and explanations for incorrect predictions (IP). The detailed analysis is shown below.

|       | Expert - GPT-3.5 | Expert - GPT-4 | GPT-3.5 - GPT-4 |
|-------|------------------|----------------|------------------|
| $R$   | 0.70             | 0.60           | 0.66             |

Table 5: Correlation coefficient between overall quality scores evaluated by expert, GPT-3.5 and GPT-4. $R$ denotes the correlation coefficient.

Participants rated our explanations with a high average score of 0.87 for overall quality, suggesting a favorable perception. As this score is closer to the maximum, it underscores the above-average quality of our explanations and highlights the efficacy of our method in clarifying the decision-making process of LLMs.

Our explanations achieved an average understandability score of 0.89, indicating a high level of clarity for participants. A variance score of 0.26 suggests consistent comprehension. However, a deeper dive reveals a disparity based on the LLM's prediction accuracy. Correct predictions (CP) had a strong average score of 0.91 and a variance of 0.26, underscoring their clarity. In contrast, incorrect predictions (IP) had a lower average of 0.74 and a variance of 0.65, suggesting they were less clear and elicited more varied responses from participants.

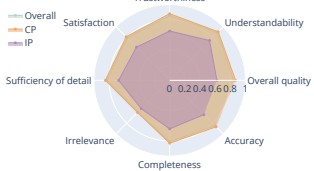

Figure 3: Evaluation on explanations, Overall, CP and IP, by humans. Note that CP aligns closely with the overall score.

Our explanations exhibited notable trustworthiness, averaging a score of 0.88 in CP. We also examined the correlation between trustworthiness and understandability. A Pearson coefficient of 0.71 indicates a strong positive relationship, suggesting that as participants better understood the explanations, their trust in the LLM's output correspondingly increased.

Participants expressed broad satisfaction with our explanations, with 86% indicating they met or surpassed expectations. A notable 97.36% of explanations were deemed to have sufficient detail. While a minor 4.54% were viewed as containing irrelevant details, the majority consensus underscored the focus and relevance of our explanations.

| | GPT-3.5 | BLOOM$_{176B}$ | GPT-NeoX | Bloomberg GPT | OPT$_{66B}$ |
|---|---|---|---|---|---|
| Acc. | 72.3% | 64.2% | 60.4% | 65.5% | 66.4% |
| Gap | 5.1% | 13.2% | 17.0% | 11.9% | 11.0% |

Table 7: The benchmark of various LLMs on CommonsenseQA test set. For ease of comparison, we include the performance gap relative to the "GPT-3.5+explanations (self-consistency)". The reported results for BLOOM, GPT-NeoX, Bloomberg GPT, and OPT are sourced from (Wu et al., 2023).

Our completeness received an average score of 0.81, suggesting overall satisfactory coverage. Notably, the median score was the maximum of 1, meaning over half of participants deemed our explanations entirely complete. This split might reflect differences in evaluators' AI backgrounds or occasional oversimplification by the model.

Our explanations achieved an accuracy score of 0.84, reflecting a positive perception. However, a deeper dive reveals a disparity: explanations for correct predictions (CP) scored 0.87, while incorrect predictions (IP) averaged 0.64. This highlights that the LLM's prediction accuracy significantly influences explanation accuracy. Furthermore, a Pearson correlation of 0.68 between accuracy and trustworthiness suggests that more accurate explanations are deemed more trustworthy.

Crowdsourcing evaluations provide a robust validation of our explanations. The positive feedback highlights the effectiveness of our approach in conveying the complexities of the LLM's decision-making in a manner that is clear, trustworthy, and satisfying to humans.

## 5.2 Objective Evaluation

We introduce $\mathbb{E}_{why}$ explanations representing the decision-making process of a LLM, as additional context to the evaluation LLMs. $\mathbb{E}_{why}$ captures the reasoning behind selecting certain answers. We conducted the evaluation using the zero-shot setting.

| Accuracy | GPT-3.5 | GPT-3.5 +explanations | Improvement | KG+LM fine-tuned (**Ours**) |
|---|---|---|---|---|
| Vanilla | 72.3% | 75.0% | 2.7% | 77.3% |
| CoT | 73.7% | 75.9% | 2.2% | —— |
| Self-consistency | 75.2% | 77.4% | 2.2% | —— |

Table 6: Comparison of accuracy of various methods w/ and w/o explanation in a QA task.

We demonstrate the results in Table 6. When evaluating the impact of incorporating explanations, we observe substantial gains in model accuracy. Specifically, we note enhancements of 2.7%, 2.2%, and 2.2% under the vanilla, CoT, and self-consistency settings respectively. This increase is an indication that the understanding of decision-making processes can be effectively transferred between language models. We also compare the evaluation model with our fine-tuned KG+LM model (as shown in Figure 1). LLMs with explanations exhibit competitive performance compared to the fine-tuned KG+LM model, even within zero-shot settings. As demonstrated in the baseline benchmark (Table 7), LLMs with explanations outperform others, with an advantage reaching up to 17%.

## 6 Conclusion

In this work, we pioneer a knowledge-enhanced method to explain the decision-making process of LLMs. This innovative approach not only generates reasoning explanations but also deepens our understanding of how LLMs operate. Based on this method, we create XplainLLM, a comprehensive dataset that comprises 12,102 commonsense questions, each accompanied by a *why-choose* and a *why-not-choose* explanation that reveals the LLM's reasoning behind its response. Furthermore, our work provides a human-centered and objective evaluation, confirming the quality and faithfulness of the generated explanations. The results demonstrate the value of our explanations in revealing the LLMs' decision-making processes.

Our work opens up new avenues for improving the explainability of LLMs and aligning their decision-making to something that is more human-understandable. We believe that XplainLLM, combined with our knowledge-enhanced approach, will prove to be a valuable resource for further research in these directions, fostering increased transparency and understanding of LLM decision-making processes.

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

## A.1  GRAPH CONSTRUCTION ALGORITHM

---

**Algorithm 1:** Graph Construction

---

**Data:** Graph $g$ with nodes $n$, input content $z$, encoding function of $\mathbb{M}$ $f_{enc}$, MLP $f_{score}$,
   Number of top nodes to select $N$

**Result:** Pruned graph $g_e$ (element-graph)

1 **begin**
2    Initialize an empty list *node_scores* ;
3    **for** *each node n in g* **do**
4       Obtain the embedding of $n$: $\mathcal{B} \leftarrow f_{enc}(n||z)$ ;
5       Compute the relevance score of $n$: $n_{score} \leftarrow sigmoid(f_{score}(\mathcal{B}))$ ;
6       Append $(n, n_{score})$ to *node_scores* ;
7    **end**
8    Sort *node_scores* in descending order based on $n_{score}$ ;
9    Select the top $N$ nodes from the *node_scores* list ;
10    Create a new graph $g_e$ with the selected $L$ nodes, preserving their edges and properties ;
11    **return** $g_e$ ;
12 **end**

---

## B.2  ADDITIONAL DETAILS OF DECISION INTERPRETATION

In section 3.1, we introduce the method for interpreting the decision-making process. We provide supplementary calculations in this section.

A $k$-layers GAT $\mathbb{G}$ is used to extract the representation from element-graph $G_e$. As introduced in the main paper, the node feature $h_t^l$ is determined as follows:

$$\boldsymbol{h}_t^l = f_e\Big( \sum_{e_s \in \mathcal{N}(e_t) \cup \{e_t\}} \alpha_{ts} m_{ts} \Big) + \boldsymbol{h}_t^{l-1}, \tag{7}$$

The $\mathbb{G}$ plays a role in updating a node feature by aggregating neighbours messages. The message $m_{ts}$ is computed according to the node properties:

$$m_{es} = f_n(h_t^l, u_t, r_{ts}), \tag{8}$$

where $f_n$ is linear transformation, $u_e$ is the one-hot vector corresponding to the type of node $e_t$, and $r_{ts}$ is the relation embedding that indicates the relation information in the edge (Yasunaga et al., 2021). $r_{ts}$ is obtained by

$$r_{ts} = f_\theta(e_{ts}, u_{ts}) = f_\theta(e_{ts}, u_t || u_s), \tag{9}$$

where $u_{ts}$ is an one-hot vector for the type of connection between $e_t$ and $e_s$, and $u_{es}$ is the concatenation of $u_t$ and $u_s$.

The attention $\alpha_{ts}$ of node $e_t$ is computed by a query vector $\mathbf{q}_e$ and a key vector $\mathbf{k}_e$,

$$\alpha_{ts} = softmax(e_t) = \frac{exp(e_t)}{\sum_{v \in \mathcal{N}(e_t)} exp(e_{tv})} = softmax\left( \frac{\mathbf{q}_t^\top \mathbf{k}_s}{\sqrt{D}} \right) \tag{10}$$

where $D$ is the feature dimension. $\mathbf{q}_e$ and $\mathbf{k}_e$ are computes by:

$$\mathbf{q}_t = f_q(h_t^l, u_t, n_{score}), \mathbf{k}_s = f_k(h_t^l, u_t, n_{score}, r_{ts}), \tag{11}$$

where $f_q$ and $f_k$ are linear transformations, and $n_{score}$ represents the *relevance score* as computed in Algorithm 1.

## C.3  EXPLANATION STATISTICS

Figure 4 is a word cloud showing the most frequently appearing words in the *why-choose* explanations. From this figure, we have a clear indication that *why-choose* explanations focus on explaining, comprehension, and interpreting predictions made by the target model.

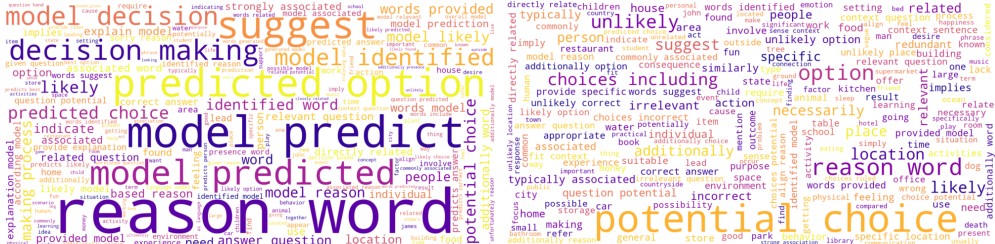

Figure 4: *why-choose* explanations.          Figure 5: *why-not-choose* explanations.

Figure 5 presents a word cloud for *why-not-choose* explanations. We note that these explanations outline the reasons behind the non-selection of specific options as predicted answers. Furthermore, *why-not-choose* explanations emphasize how the target model determines the likelihood of different answer choices. We also observe that the target model handles a wide array of topics, which can be crucial components in the "why not" reasoning process.

## D.4 INSTANCE EXAMPLE

We present an example from our dataset to illustrate the data instance.

---
**Explanation Example**

**Question:** The people danced to the music, what was the music like for them?
**Answers:** soothing, vocal or instrumental, loud, universal, enjoyable
**Label:** enjoyable
**Predicted Label Index:** 4
**Label Matched:** true
**Reason-elements:** play_mozart, listening_to_classical_music, together, abnd, dancefest, irritation, ...
**Top-k Reason-elements:** play_mozart, listening_to_classical_music, together, abnd, dancefest
**Explanation (Why):** Since the words "play_mozart" and "listening_to_classical_music" suggest that the music was likely of high quality and enjoyable to listen to, and since the words "together" and "dancefest" suggest that the people were likely in a social and festive setting, the model predicts that the music was enjoyable for them. Therefore, the predicted option is "enjoyable".
**Explanation (Why-Not):** The other potential choices can be combined into three categories: 1) "soothing" and "vocal or instrumental" do not match with the festive and social setting suggested by the words "together" and "dancefest"; 2) "loud" is not necessarily correlated with enjoyment and could be unpleasant for some people; 3) "universal" does not give any indication of the quality or character of the music, and is therefore too vague to be a valid choice.

---

The format of our dataset is as follows:

---
**Data Schema Description**

**question:** typeof(string)
**answers:** typeof(list_of_strings)
**label:** typeof(string)
**predicted_label:** typeof(string)
**label_matched:** typeof(boolean)
**concept:** typeof(list_of_strings)
**topk:** typeof(list_of_strings)
**explanation_why:** typeof(string)
**explanation_why_not:** typeof(string)

---

## E.5 EXPERIMENTS

In this section, we describe the details of our evaluation that were omitted in Section 4 due to space constraints.

### E.5.1 MODEL PARAMETERS

To train our $\mathbb{M}$, we use a dropout rate of 0.2, a batch size of 64, and a learning rate of 1e-5, optimized with RAdam. The model is fine-tuned on a single NVIDIA A100 GPU for approximately 3 hours. Our $g_k$ containing 799,273 nodes and 2,487,810 edges. Our $g_e$ is pruned based on $g_k$ to retain 200 high-ranking nodes with a hop size of 2. The $\mathbb{G}$, specifically, consists of 200 dimensions and 5 layers. The learning rate in our experiments is 1e-3.

For GPT-3.5 and GPT-4, we set the temperature, the frequency penalty and the presence penalty to be 0.0, and the top probability to be 1.0. All experiments involving GPT-3.5/4 are conducted through the available online API. The example of in-context learning in the CoT setting (i.e. Table 6) follows the original work (Lu et al., 2022). For self-consistency setting in Table 6, we sampled 5 possible answers for each question and picked the most likely one.

### E.5.2 DATA COLLECTION

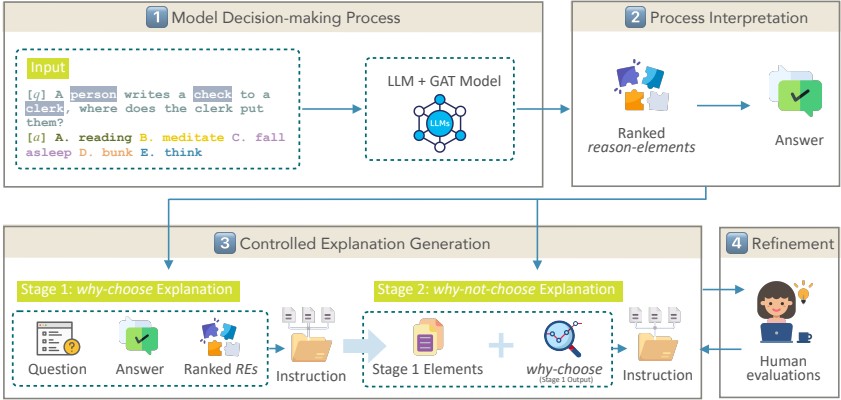

Figure 6: Data Collection Process.

Figure 6 shows the process of data collection:

(1) Given a question, we retrieve its relevant knowledge using the KG. The retrieved graph is then pruned based on scores influenced by the LLM, resulting in what we term the *element-graph*. The *element-graph* is processed by a specialized GAT model (known formally as Decision Interpretation). Leveraging attention mechanisms, we obtain the essential representations for interpretation.
(2) The model's decision-making is interpreted through the ranked *reason-elements* and the predicted answer.
(3) A controllable dual-stage process generates the explanations: Stage 1: The initial phase focuses on generating the "why-choose" explanations. Stage 2: Building upon the outputs and elements of Stage 1, we then generate the "why-not-choose" explanations.
(4) We conduct a human evaluation to identify errors in the explanations. If discrepancies arise, explanations are reverted to Step 3 for refinement. This process not only helps prevent potential issues arising from bad explanations generated by the LLM but also maintains human-aligned clarity and relevance.

### E.5.3 INSTRUCTIONS FOR EXPLANATION GENERATION

The generator model $\mathbb{F}$ generates *why-choose* and *why-not-choose* explanations of the LLM $\mathbb{M}$'s behavior based on a predefined set of instructions.

Recall that explanations are generated using instructions of the form

$$\textbf{(1)} \ W : [Z], \ P : [y], \ T : [O], \ C : [\{R|R \in E\}]; \ \textbf{(2)} \ I : [\mathbb{E}_{why}], \ \hat{T} : [O], \tag{12}$$

Specifically, the instruction symbols $W$, $P$, $T$, $O$, $C$, $I$ and $\hat{T}$ have the following meanings:

1. $W$ represents "The question is ..."

2. $P$ represents "The predicted choice is ..."

3. $T$ represents "According to the model top reason-elements, explain the model reasoning process using ..."

4. $O$ represents $\langle since..., ....\rangle$

5. $C$ represents "The top reason-elements are".

6. $I$ represents "According to ..."

7. $\hat{T}$ represents "Explain why the model doesn't choose other answers ..."

## F.6 EVALUATION MATERIALS

### F.6.1 QUESTIONS AND EVALUATION INSTRUCTIONS

For each QAE triple, we provide eight questions for evaluators. Each question includes three score levels: 1 for disagree, 2 for neutral, and 3 for agree. The questions and instructions in our evaluation are as follows:

---

**Q0: This is a good explanation**

| | |
|---|---|
| 1 | (Disagree) The explanation is illogical or inconsistent with the question and/or does not adequately cover the answer choices. |
| 2 | (Neutral) The explanation is somewhat logical and consistent with the question but might miss some aspects of the answer choices. |
| 3 | (Agree) The explanation is logical, consistent with the question, and adequately covers the answer choices. |

**Q1: I understand this explanation of how the AI model works.**

| | |
|---|---|
| 1 | (Disagree) The explanation is unclear or contains overly complex terms or convoluted sentences. |
| 2 | (Neutral) The explanation is somewhat understandable but might contain complex terms or convoluted sentences. |
| 3 | (Agree) The explanation is clear, concise, and easy to understand. |

**Q2: I trust this explanation of how the AI model works.**

| | |
|---|---|
| 1 | (Disagree) The explanation is unclear or contains overly complex terms or convoluted sentences. |
| 2 | (Neutral) The explanation is somewhat credible but contains some elements that I find doubtful or questionable. |
| 3 | (Agree) The explanation is credible and aligns with my understanding of how AI models work. |

**Q3: This explanation of how the AI model works is satisfying.**

| | |
|---|---|
| 1 | (Disagree) The explanation does not meet my expectations and leaves many questions unanswered. |
| 2 | (Neutral) The explanation somewhat meets my expectations but leaves some questions unanswered. |
| 3 | (Agree) The explanation meets my expectations and satisfies my query. |

**Q4: This explanation of how the AI model works has sufficient detail.**

| | |
|---|---|
| 1 | (Disagree) The explanation lacks detail and does not adequately cover the AI model's decision-making. |
| 2 | (Neutral) The explanation provides some detail but lacks thoroughness in covering the AI model's decision-making. |
| 3 | (Agree) The explanation is thorough and covers all aspects of the AI model's decision-making. |

**Q5: This explanation of how the AI model works contains irrelevant details.**

| | |
|---|---|
| 1 | (Disagree) The explanation does not contain any irrelevant details. |
| 2 | (Neutral) The explanation contains some irrelevant details. |
| 3 | (Agree) The explanation contains many irrelevant details. |

**Q6: This explanation of how the AI model works seems complete.**

| | |
|---|---|
| 1 | (Disagree) The explanation does not adequately cover the answer choices and leaves many aspects unexplained. |
| 2 | (Neutral) The explanation covers most answer choices but leaves some aspects unexplained. |
| 3 | (Agree) The explanation covers all answer choices and leaves no aspect unexplained. |

**Q7: This explanation of how the AI model works is accurate.**

| | |
|---|---|
| 1 | (Disagree) The explanation does not accurately reflect the AI model's decision-making. |
| 2 | (Neutral) The explanation somewhat reflects the AI model's decision-making but contains some inaccuracies. |
| 3 | (Agree) The explanation accurately reflects the AI model's decision-making. |

---

### F.6.2 DETAILS OF HUMAN-CENTERED METRICS

The meaning of metrics used in the human-centered evaluation are as follows:

1. **Overall quality** reflects the overall effectiveness of explainability. It reveals how effectively explanations convey the decision-making process of the AI models to the human users.

2. **Understandability** evaluates how well a human can comprehend the model's output and explanations. It captures the clarity and coherence of the generated text.

3. **Trustworthiness** measures the human evaluator's confidence in the model's outputs and explanations. It evaluates whether the explanations appear reliable, credible, and based on sound reasoning.

4. **Satisfaction** captures the overall contentment of the evaluator with the explanations. It measures whether the outputs meet the evaluator's needs and expectations in terms of quality, relevance, and utility.

5. **Sufficiency of detail** evaluates whether the explanations provide a sufficient level of detail. It evaluates whether the responses are adequately descriptive and provide all necessary information to fully answer the question or task.

6. **Irrelevance** evaluates whether the explanations include any unnecessary or irrelevant information.

7. **Completeness** measures whether the explanations address the decision behaviors of the model.

8. While we also measure **accuracy** objectively, the human evaluation of accuracy assesses whether the explanations align with the evaluator's knowledge or expectations. It measures whether the explanations can reflect if the model's outputs are factually correct and contextually appropriate.

