# OpenReview forum: "XplainLLM: A QA Explanation Dataset for Understanding LLM Decision-Making"
_ICLR.cc/2024/Conference — Submitted to ICLR 2024_

### Official Review · Reviewer_MEtr · 2023-10-26

**Soundness:** 2 fair
**Presentation:** 2 fair
**Contribution:** 2 fair
**Rating:** 3
**Confidence:** 4

**Summary:**

The paper's primary goal is to enhance the interpretability and explainability of Language Learning Models (LLMs) by integrating a generated reasoning process into a QA model as input. The authors conducted experiments on a commonsense QA task, employing a Graph Attention Network (GAT) model to encode an external knowledge graph and identify the most influential nodes/keywords as reasoning evidence. Subsequently, GPT-3.5 was used to generate explanatory paragraphs, elucidating why each answer candidate is considered correct or incorrect. These generated decision-making process paragraphs were then utilized by the QA model for predictions. The experimental findings demonstrated a noteworthy improvement of 5.1% compared to GPT-3.5 on the CommonsenseQA dataset, showcasing the effectiveness of the newly-introduced decision-making process. Furthermore, a human study validated the high quality of the generated explanations. The authors will also release their dataset, XplainLLM, marking it as the first dataset designed to capture the pivotal elements influencing the model's reasoning process.

**Strengths:**

1. The proposed method, which incorporates a decision-making process into a QA model, yields significant improvements. This highlights that offering explanations in natural language not only enhances model explainability, but also boosts the performance of the QA model.
2. As a result of this work, a new dataset will be created, providing both "why" and "why not" justifications for each answer candidate. This resource could be valuable for future research within the community.

**Weaknesses:**

1. The writing requires improvement in two key areas: (1) The technical details are unclear, particularly regarding the training of the QA model M and graph model G, as well as the evaluation process for the reason-elements. (2) Numerous typos need correction. E.g., in the subsection of Decision Interpretation on Page 4, "consider any node..." needs to be "for any node".

2. The task 2 setting is a bit redundant. Since one of the input E_why already contains the predicted answer, why we need to feed it to another QA model for prediction? How often does the QA model's predicted answer differ from E_why?

3. It's important to acknowledge that the explanations generated by the model may be prone to noise, as they are produced by an automated generator rather than a human. While efforts have been made to evaluate both "why" and "why not" explanations from various perspectives, their quality remains, the quality is still questionable. Notably, only the generated explanations are assessed, while the selected reason-elements from GAT are neither evaluated nor supervised. If these reason-elements are incorrect or suboptimal, it may impact the overall quality of the resulting natural language explanations.

**Questions:**

1.  What is "||" operator on page 4?

---

> ### Author Response · Authors · 2023-11-21
>
> Thank you for taking the time to review our paper and for the constructive feedback. We appreciate the opportunity to address your concerns:
>
>
>
>
> **Response to Weaknesses:**
>
> **1. (W1.1) Clarification on Technical Details:**
>
> 1. Training of Model M and Model G: Model M is our pre-trained LLM, and Model G is GAT to extract the reason-elements. We use the pre-trained RoBERTa-large as Model M, and the GAT is trained with the Model M (embedding concatenation). They trained with the loss function of cross-entropy loss. The optimization involves fine-tuning both the knowledge representation (via model G) and the predictive components (via model M). The training details can be found in Appendix E.5.1-2. We will revise Section 3.1 to clarify the training details.
> 2. Evaluation Reason-elements: The reason-elements are
> the key elements that contribute to the decision-making process of the model. They are extracted from the graph modular, mainly based on the attention weights of the GAT. Therefore, they are evaluated by the accuracy of the decision model. We will revise Section 3.1 to clarify the evaluation of reason-elements.
>
> **2. (W1.2) Correction of Typos:**
> We appreciate the reviewer pointing out the typos in our manuscript. We will thoroughly review the document to correct these errors, and ensure that our revised version is free of typos.
>
> **3. (W2) Clarification on Model Selection:**
>
> The design of Task 2 is primarily focused on exploring the transferability of learned knowledge of one LLM to enhance the performance of another LM. The key points are as follows:
> - **Purpose of E_why:** The E_why serves as a knowledge transfer medium. It encapsulates the reasoning used by the first LLM to arrive at its prediction.
> - **Transferability via E_why:** Feeding E_why to another LLM is not about re-predicting the answer but rather about examining how the explanation influences the understanding and performance of the second LLM. This process tests the effectiveness of the explanation in conveying the reasoning knowledge of the first LLM to another. We investigate the potential of explanations to serve as a knowledge bridge between different models here.
> - **Impact on Performance:** Our experiments demonstrate that by utilizing the explanations generated by one LLM, another LLM can enhance its performance (one example is shown below). With the help of explanations, the second LLM increases its accuracy by 2.7%. The different ratio of predicted answers after transferring is around 2%-4%.
> The results indicate LLMs can benefit from the reasoning and knowledge encapsulated in the explanations, leading to improved decision-making and understanding.
>
>
>     | Model                      | Method    | Accuracy |
>     |----------------------------------|----------|----------|
>     | GPT-3.5                          | Vanilla  | 72.3%    |
>     |GPT-3.5 +explanations| Vanilla  | 75.0%    |
>
>
> **4. (W3) Clarification on Dataset Quality:**
> We acknowledge there are limitations in the current design and we have discussed them in the limitation subsection. The quality of KG (which is ConceptNet in our case) and more advanced GNN are important factors that can affect the effectiveness of our method.
>
> Our task is to explain the reasoning process of LLMs. We are different from the typical explanation generation task, which is to generate truthful explanations of certain questions. Such tasks might rely on humans to write explanations for the questions. In our explanation task, we need to extract the reasoning process from the decision model, and "translate" it into human-understandable language. Thus, we use an automatic way to generate explanations.
>
> To ensure the faithfulness of the explanations, we use the key reason-elements extracted from the decision model as the prompt to the generator model. The generator model is more like a "translator" that translates the decision model's reasoning into human-understandable language. The reason-elements are not independent from the decision model. They are extracted based on the attention weights of the GAT. Therefore, they are evaluated by the accuracy of the decision model.
>
>
> We conducted comprehensive experiments to validate the faithfulness of our explanations. Actually, we have included the reason-elements in the evaluation. We will add more details in the revised version.
> The evaluation included expert evaluation, human evaluation and automated evaluation to assess the quality of our explanations. Our results show that our explanations are with high quality and accuracy. To reduce the bias, we followed the methodology outlined by Hoffman et al. (2018) [1] for cross-human alignment and evaluation questions design. This approach helps to minimize bias and ensure the faithfulness of our explanations.
>
>
> ---
> [1] Hoffman, Robert R., et al. "Metrics for explainable AI: Challenges and prospects." arXiv preprint arXiv:1812.04608 (2018).
>
> **Not yet finished, continue to the next comment...**

---

> > ### Author Response · Authors · 2023-11-21
> >
> > **Response to Questions:**
> >
> > **1. (Q1) Clarification on Notation:**
> >
> > The "||" operator mentioned on page 4 represents concatenation. We will clarify "||" in our revised version. We will ensure consistency and avoid any confusion regarding the notation used in our paper.
> >
> > ---
> > We are grateful for the opportunity to clarify our work. We hope that our response has addressed your concerns. We look forward to hearing from you.

---

> > ### Comment · Reviewer_MEtr · 2023-11-23
> >
> > Thank you for providing detailed clarifications. However, I remain unconvinced about the dataset's quality and the rationale for using the XplainLLM dataset in future research.
> >
> > While the reason-elements are extracted based on GAT attention weights, the lack of supervision by ground truth during Model G's training raises concerns about the reliability of these weights. The fact that utilizing explanations in a second LLM yields gains indicates their potential usefulness but doesn't necessarily reflect high-quality. Fundamentally, they appear too noisy to be considered reliable and could be easily replaced by those generated from a more advanced LLM, serving as a substitute for GAT.
> >
> > A suggestion is to demonstrate the transferability of reason-elements directly, rather than relying on E_why. E_why, being the output of two stacked "black boxes" (GAT + LLM), primarily serves a human audience and introduces unnecessary noise when assessing the true value of the machine reasoning process (i.e., reason-elements).

---

> > > ### Author Response · Authors · 2023-11-23
> > >
> > > Dear Reviewer,
> > >
> > > Thank you for your reply. We understand and appreciate your concern. We would like to clarify that our work is to explain the reasoning process of a decision model.
> > >
> > > The model G is fine-tuned with the decision model and is supervised by the ground truth, which is a joint training. Thus, it can reflect the reasoning process of the decision model. The quality of the explanations is evaluated by humans and model performance. Our decision model can achieve very high performance **(77.3%)**, and better than GPT-3.5 **(72.3%)** [table 6 and 7, page 9]. (Our decision model did not include the explanations or any capability of GPT.) The second LLM here (generator model) is like a "translator" that translates the decision model's reasoning process into human-understandable explanations.
> > > We understand that the reviewer's workload is heavy, so you might skip the details in the methodology section. We describe the learning process in Section 3.1 (Subsection **Decision Interpretation**). Please refer to it for more details.
> > >
> > >
> > > The main advantage of our approach is that it transforms the 'black-box' nature of LLMs (**decision model**) into a more transparent and interpretable structure. While GATs can be complex, they are not entirely black-box models. They offer a level of interpretability by explicitly modeling the relationships and relative importance between nodes in a graph. This allows us to trace back and understand why certain nodes are given more prominence in the reasoning process.
> > >
> > > As we mentioned in previous rebuttal, we conducted comprehensive experiments to validate the faithfulness of our explanations.
> > > We included **expert evaluation, human evaluation and automated evaluation** to assess the quality of our explanations. Our results show that our explanations are with **high quality and accuracy** (Table 4, page 8). In XAI, the main goal is to make the reasoning process of machine models more transparent, interpretable, and human-understandable. Our work is in this direction. Our first job is not to improve the performance of the decision model, but to explain the reasoning process of the decision model to human.
> > >
> > > We believe our dataset is broader applicability beyond XAI: Our dataset is not limited to evaluating XAI models. It has broader applications, such as:
> > > 1. Debugging the model: Our dataset can be used to debug a model by identifying the reason-elements that are not aligned with human understanding and expectations.
> > > 2. Training and fine-tuning the model: Our dataset can be used to train and fine-tune the model to generate more accurate and interpretable explanations.
> > > 3. As a benchmark for future research: Our dataset can be used as a benchmark for future research in XAI. It can be used to evaluate the performance of different XAI models and compare their effectiveness.
> > >
> > > We hope this explanation addresses your concerns, and we are grateful for your feedback. Please let us know if you have any further questions or comments, we are happy to address them.
> > >
> > > \
> > > Sincerely,
> > >
> > > Authors

---

> > > > ### Author Response · Authors · 2023-11-23
> > > >
> > > > Dear Reviewer,
> > > >
> > > > We have released our dataset (http://anonymous.4open.science/r/XplainLLM), and we warmly welcome you to check the quality of our dataset. We also randomly selected an example in Appendix D.4 (page 15), please refer to it for more details.
> > > >
> > > > We believe our dataset makes a substantial contribution to the field of XAI by addressing a critical need for transparency and explainability in AI decision-making, and enabling general users to easily understand this process, which can be helpful for future human-centered applications and more.
> > > >
> > > > \
> > > > Sincerely,
> > > >
> > > > Authors

---

> > > > > ### Author Response · Authors · 2023-11-23
> > > > >
> > > > > Dear Reviewer,
> > > > >
> > > > > To help you better understand the value of our dataset, we would like to introduce more recent works in this area ([1], [2], [3]).
> > > > >
> > > > > Here is an example of [1] (a dataset) from OpenAI, explaining why word "dollars" is generated (a decision-making process) by GPT-2:
> > > > > ```
> > > > > A broad effort is under way to understand what really works in health care, perhaps leading to better value for dollars spent.
> > > > >
> > > > > Explanation:
> > > > > Token: dollars
> > > > > mentions of "American" and related terms
> > > > > ```
> > > > >
> > > > > The weakness of [1] is that it can only explain a single word not complete reasoning, and it is still difficult for humans to understand the reasoning process of the LM. This dataset is also **automatically generated**, and not human-annotated.
> > > > >
> > > > > We want to emphasize our novelty: (1) our dataset provides a new way of explaining how LLMs work (via more transparent surrogate components, e.g., reason-elements), and (2) explanations of LLMs' decision-making are more human-understandable compared to other works in this area.
> > > > >
> > > > > We hope this additional explanation addresses your concerns, and we are grateful for your feedback.
> > > > >
> > > > > \
> > > > > Sincerely,
> > > > >
> > > > > Authors
> > > > >
> > > > > ---
> > > > > [1] Steven Bills, Nick Cammarata, Dan Mossing, Henk Tillman, Leo Gao, Gabriel Goh, Ilya Sutskever, Jan Leike, Jeff Wu, and William Saunders. Language models can explain neurons in language models. https://openaipublic.blob.core.windows.net/neuron-explainer/ paper/index.html, 2023.
> > > > >
> > > > > [2] Weld, Daniel S., and Gagan Bansal. "The challenge of crafting intelligible intelligence." Communications of the ACM 62, no. 6 (2019): 70-79.
> > > > >
> > > > > [3] Lai, Vivian, Han Liu, and Chenhao Tan. "" Why is' Chicago'deceptive?" Towards Building Model-Driven Tutorials for Humans." In Proceedings of the 2020 CHI Conference on Human Factors in Computing Systems, pp. 1-13. 2020.

---

> ### Author Response · Authors · 2023-11-22
> **Look Forward to Your Response**
>
> Dear Reviewer,
>
> As we approach the conclusion of the author-reviewer discussion period, we eagerly anticipate your thoughts on our rebuttal. We have endeavored to comprehensively address the concerns you raised. If there are any remaining issues or queries, we warmly welcome your input.
>
> We appreciate your feedback and look forward to your response.
>
> \
> Best regards,
>
> Authors

---

### Official Review · Reviewer_ZCRc · 2023-11-01

**Soundness:** 3 good
**Presentation:** 3 good
**Contribution:** 3 good
**Rating:** 6
**Confidence:** 3

**Summary:**

This paper introduces a dataset, XplainLLM, consisting of explanations that link the LLM reasoning process to the entities and relations within knowledge graphs (KGs). The explanations are presented in a human-understandable way, including why-choose and why-not-choose explanations. XplainLLM contains 12,102 question-answer-explanation (QAE) triples. The explanations effectively enhance the performance of LLMs on commonsense QA. Both human and automatic evaluations are conducted to evaluate the quality of explanations.

**Strengths:**

-	Novelty. This paper proposes a novel method that incorporates highlighted elements and guided instruction to generate a free-text explanation. KGs and graph attention networks (GAT) are leveraged to extract reason elements w.r.t. model decision-making. The reason elements are then converted into human-understandable explanations.
-	This paper is well-written and easy to follow. The experiments are comprehensive and convincing, showing the quality of collected explanations through both human and automatic evaluations and the utility of explanations in enhancing the performance of LLMs.
-	The dataset is released, which may benefit future research.

**Weaknesses:**

-	It is not clear why the RoBERTa-Large model is used as the decision model. Since the final explanations are generated by the GPT-3.5-turbo model, the performance gain may benefit from the external knowledge provided by GPT-3.5-turbo. It is worth considering different types of decision and generator models.
-	The proposed framework seems promising in explaining a model’s decision-making in a human-understandable way. However, it is not clear how much it benefits the dataset construction. Solely using GPT-3.5-turbo or GPT-4 for prompting can facilitate the creation of a dataset with question-answer-explanation triples.

**Questions:**

-	Why is RoBERTa-Large used as the decision model instead of more advanced LLMs?
-	To what extent the reason-elements extraction with KGs and GAT can enhance the explanation generated by GPT-3.5-turbo? How about solely using GPT-3.5-turbo to generate explanations?

---

> ### Author Response · Authors · 2023-11-21
>
> Thank you for taking the time to review our paper and for the constructive feedback. We appreciate the opportunity to address your concerns:
>
> **Response to Weaknesses:**
>
> **1. (W1) Clarification on Model Selection:**
>
> **a. We selected RoBERTa-Large for several reasons:**
>    - RoBERTa-Large has been widely used in the NLP community and is well-known for its robust performance. Its widespread use in the NLP community makes it a reliable base model for evaluating new methods and approaches. Our explanation quality can be influenced by the performance of the decision model, so we choose a model with good performance as the decision model.
>    - Our dataset focuses on explaining the reasoning process of LLMs. There is no human-understandable explanation to demystify the reasoning process of LLMs. RoBERTa is a fundamental model in the LLM family, we believe that it is important to start with such a model to establish a baseline for future research.
>
> **b. Explanation Generation:**
> - We chose GPT-3.5 as the generator model because of its advanced language capabilities. While it's true that GPT-3.5-turbo's internal knowledge might contribute to the performance gain, our design tries to minimize its influence. The main advantage of our approach is that it transforms the 'black-box' nature of LLMs into a more transparent and interpretable structure. With our graph modular design, we narrow down the knowledge search space of the LLM, providing a clearer understanding of the model's decision-making process. By using strong constrained instruction and the reason-elements learned from the decision model as our prompt to generator model, we can control the generation content. The generator model is more like a "translator" that translates the model's reasoning into natural, human-understandable language.
>
> As a pioneer study, we believe that our dataset can be further improved and expanded. We will continue to expand our dataset to include a variety of LLMs in our future work. We hope that our work can inspire more research in this area and contribute to the development of XAI, making it more interpretable, trustworthy and human-centered.
>
> **2. (W2) Clarification on Our Task:**
> While GPT-3.5 or GPT-4 are powerful in generating human-like text, they are not designed to explain the reasoning process of curtain LLMs. The reviewer mentioned question-answer-explanation triples, such task is different from our task. Our task is to explain the reasoning process of curtain LLMs, which is different from the generation task. Without the knowledge of the decision model, the generator model cannot generate explanations that reflect the reasoning process of the decision model.
>
> Our approach can interpret the reasoning process, and link the reasoning process to a generator model. As mentioned before, the generator model is more like a "translator" that translates the decision model's reasoning into human-understandable language.
>
> ---
> **Response to Questions:**
>
> **1. (Q1) Model Selection:**
> First, please refer to the response to **W1.1** (*We selected RoBERTa-Large for several reasons*).
>
> It is important to understand the reasoning process of more advanced LLMs. However, we find there is no human-understandable explanation for LLMs, even for "small"/"not such advanced" LLMs. Our work is a pioneer study in this area. Furthermore, our approach is not limited to RoBERTa-Large. It can be easily adapted to other LLMs. We will keep working on expanding our dataset to include a variety of LLMs in our future work.
>
>
> **2. (Q2) Clarification on Explanation:**
>
> First, please refer to the response to **W2** (*Clarification on Our Task*).
>
> Since our task is to explain the reasoning process of curtain LLMs, we need to extract the learning knowledge from the decision model to generate explanations. The use of KGs and GAT for extracting reason-elements ensures that the explanations can reflect the reasoning process of the decision model. KGs provide structured and domain-specific knowledge, which when combined with the GAT, allows us to extract the key reason-elements that contribute to the decision-making process. There are two main advantages of our approach: (1) enhancing the explainability of the decision model's outputs, and (2) reducing the irrelevant knowledge and improving the performance of the decision model.
>
> We want to emphasize that our methodology is universally applicable across a diverse range of LLMs. The foundational elements of our approach, such as the integration of KGs and the interpreting component, are designed with flexibility in mind and are not specific to RoBERTa. These components can be adaptable to the architectures and functionalities of various LLMs.
>
> ---
> We are grateful for the opportunity to clarify our work. We hope that our response has addressed your concerns. We look forward to hearing from you.

---

> ### Author Response · Authors · 2023-11-22
> **Look Forward to Your Response**
>
> Dear Reviewer,
>
> As we approach the conclusion of the author-reviewer discussion period, we eagerly anticipate your thoughts on our rebuttal. We have endeavored to comprehensively address the concerns you raised. If there are any remaining issues or queries, we warmly welcome your input.
>
> We appreciate your feedback and look forward to your response.
>
> \
> Best regards,
>
> Authors

---

### Official Review · Reviewer_w11A · 2023-11-07

**Soundness:** 2 fair
**Presentation:** 2 fair
**Contribution:** 2 fair
**Rating:** 3
**Confidence:** 3

**Summary:**

This paper introduces a dataset for helping humans understand the decision-making process of LLMs by integrating the explanations to entities and relations present in a knowledge graph (KG). Operating in a multiple-choice setting for question answering, the dataset captures explanations for both the “why” as well as the “why-not” (i.e. explanations for options that are incorrect).

The paper first constructs a pruned knowledge graph starting from the entities present in the question and answer options. The pruning is done by scoring nodes and edges wrt the question and it follows the steps of previous work (Yasunaga et al 2021). After forming the question subgraph, the model employs graph attention networks (GAT) on the subgraph to form node embeddings of entities. The top-m nodes in the graph w.r.t the attention scores are selected as the \textit{reason-elements}.

Using the \textit{reason-elements} and the correct answer, explanations in natural language were generated by GPT-3.5. Explanation are generated for supporting the correct answer option as well as for not supporting the incorrect options, all while being conditioned on the reason-set. The quality of the dataset is verified by humans as well as via automated models (GPT-3.5 and GPT-4).

**Strengths:**

**Originality**

The GAT set up closely follows Yasunaga et al 2021, however the current paper uses the top-attended nodes from the output of the GAT model as anchor points (reason-set) to generate a dataset with explanations. Even though I am not convinced about the quality of the reason-set, this part of the paper is novel

**Quality**

I am not convinced about the quality of the dataset produced as I expand in the next section. Hence I am not sure about the current quality of the paper.

**Clarity**

Overall, the paper was not difficult to follow, however a lot of important details have been put into the appendix which, sometimes, affect the readability. For instance, at the end of Sec 3.1, a lot of new notations have  been introduced (e,g, P, T, C) and the readers have been asked to refer to the appendix. Therefore I believe the paper will benefit from a round of re-writing.

**Significance**

Given that I am not sure about the quality of the dataset as a benchmark for explainable AI, I am not sure the paper in its current form would be significant for the XAI community.

**Weaknesses:**

* The biggest confusion I had while reading the paper is whether the paper introduces a model for explainable AI or introduces a dataset for XAI. The paper claims that it does the latter, however the dataset is automatically derived from the outputs of a model itself. For example, the reason set is the output of a graph neural network model which itself is a black-box and can be inaccurate. What is the guarantee that the reason-set faithfully reflects the working of the GAT model and therefore how can we be convinced that a dataset derived from a model can be used to evaluate other XAI models in an unbiased way?
  * A related point is what is the guarantee that the dataset will not favor models that are not from the same family as the model used to create the dataset? What is the guarantee that it wont penalize a model which in-reality is a better XAI model but produces explanation which are different from the current dataset.

* The reason elements are selected as the top-k elements based on attention scores. However the attention scores are itself computed by a black-box graph neural network model. What is the guarantee that the reason-elements are faithful and reflect the actual decision making process of the model?
  * For example, in Figure 1, the reason elements selected by the model (e.g. delay, delivery, maintain, etc) look arbitrary to me given the question. Since these form the back-bone of the explanation generation process, it is important that the reason-elements truly reflect the model decision-making process and I am not convinced that is the case.
  * A similar observation about the example in appendix D.4. For the given question: “The people danced to the music, what was the music like for them?” - why would a top-reason element be “play_mozart”? Introducing Mozart for this question is unnecessary and might make the explanation meaningless. Also, if used as a benchmark, if a model does not generate Mozart as a part of its explanation, why should it be penalized?
* The explanation generated from the reason-elements also seem arbitray to me. Even though all the reason-elements are covered in the explanation, the provided explanation does not seem sound to me. For example, “”the keywords maintain and cease suggest that the clerk wants to keep the check in a secure location”. Why is this a valid explanation generated by the LLM?
  * I have similar observation and comments for the explanation generated in Why-not-choose column. For example, why would a desk-drawer have too many potential locations to search through? This seems to be model hallucinations to justify the correct answer.
* The paper claims “Aligning Human Understanding and Model Explainability” as one of the major contributions in the intro section and it claims that this explanation can be used to train RLHF models. While I can believe that would be the case, I did not find any experiments to support this. Since this is mentioned as a core-contribution, I believe this merits validation by experiments.

**Questions:**

* My biggest question is the effectiveness of the reason-set as an output of the GAT process? How accurate and effective are they? Because from the examples given in Fig 1 and appendix, they are unfortunately not very effective.

* Instead of presenting this paper as a dataset generation paper, why not present this model + explanation from GPT-3.5 as a model for XAI? I believe the model itself has merits and analyzing its results could be interesting.

---

> ### Author Response · Authors · 2023-11-21
>
> Thank you for taking the time to review our paper and for the constructive feedback. We appreciate the opportunity to address your concerns:
>
>
>
> **Response to Weaknesses:**
>
> **1. (W1) Clarification on Dataset and Faithfulness:**
>
> Our paper primarily introduces a dataset for XAI, and here's how we ensure its relevance and utility:
>
> - **Dataset Construction:** The core contribution of our work is the development of a novel dataset for XAI. The dataset is designed to introduce a way to dive deep into understanding the decision-making of LLMs, transforming a black box into human-understandable explanations. Moreover, our dataset can be helpful for future research in XAI, for example, debugging a model, evaluating its performance, designing human-AI interactions, etc.
> - **Clarification on Reason-elements:** The dataset is derived from certain outputs of models. The reason-elements generated by the GAT are not used directly as the ground truth for explainability. Instead, it serves as a starting point for generating more refined and human-centered explanations.
> - **Clarification on Faithfulness:**
>   - While GATs can be complex, they are not entirely black-box models. They offer a level of interpretability by explicitly modeling the relationships and relative importance between nodes in a graph. This allows us to trace back and understand why certain nodes are given more prominence in the reasoning process.
>   - The main advantage of our approach is that it transforms the 'black-box' nature of LLMs into a more transparent and interpretable structure. With our graph modular design, we narrow down the knowledge search space of the LLM, providing a clearer understanding of the model's decision-making process.
>   - We conducted comprehensive experiments to validate the faithfulness of our explanations. We included expert evaluation, human evaluation and automated evaluation to assess the quality of our explanations. Our results show that our explanations are with high quality and accuracy. To reduce the bias, we followed the methodology outlined by Hoffman et al. (2018) [1] for cross-human alignment and evaluation questions design. This approach helps to minimize bias and ensure the faithfulness of our explanations.
> - **Broader Applicability Beyond XAI Model Evaluation:** Our dataset is not limited to evaluating XAI models. It has broader applications, such as:
>   - **Debugging the model:** Our dataset can be used to debug a model by identifying the reason-elements that are not aligned with human understanding and expectations.
>   - **Training and fine-tuning the model:** Our dataset can be used to train and fine-tune the model to generate more accurate and interpretable explanations.
>   - **As a benchmark for future research:** Our dataset can be used as a benchmark for future research in XAI. It can be used to evaluate the performance of different XAI models and compare their effectiveness.
>
> As a pioneer study, we believe that our dataset can be further improved and expanded. We hope that our work can inspire more research in this area and contribute to the development of XAI, making it more interpretable, trustworthy and human-centered.
>
> ---
> [1] Hoffman, Robert R., et al. "Metrics for explainable AI: Challenges and prospects." arXiv preprint arXiv:1812.04608 (2018).
>
> **Not yet finished, continue to the next comment...**

---

> > ### Author Response · Authors · 2023-11-21
> >
> > **2. (W2) Clarification on Faithfulness:**
> > [Similar comment to W1]
> > This concern is similar to W1. As we have mentioned in W1, our design transforms the 'black-box' nature of LLMs into a more transparent and interpretable structure. With our graph modular design, we narrow down the knowledge search space of the LLM, providing a clearer understanding of the model's decision-making process. The attention mechanism is used to identify the most relevant nodes (reason-elements). The attention weights are not arbitrary, but are the result of a learned process. It reflects the decision-making process of the model.
> > - **(W2.1) Faithfulness of Reason-elements:** The top-k reason-elements are selected based on the attention weights. These attentions are the most influential in the decision-making process of the model. The faithfulness of these reason-elements is grounded in the GAT's ability to accurately capture and identify the most relevant knowledge in the graph. We acknowledge that the GAT is not perfect, and there is room for improvement. However, we believe that our approach is a step in the right direction towards making LLMs more interpretable and trustworthy.
> > - **(W2.2) Contextual Relevance and Model Limitations:** In the examples you mentioned, such as the inclusion of "play_mozart" for a question about music, it's important to recognize that models may sometimes establish connections that, while potentially relevant, might not align with every human interpretation. This is a limitation inherent in current AI models, stemming from their reliance on patterns found in training data. The model's decision to include such elements is based on its learned associations and not on arbitrary selection. While this may not always align with human expectations, it is a reflection of the model's reasoning process. We have added a related analysis in the limitation section.
> >
> >     We also want to emphasize that our dataset is not limited to be a benchmark for XAI models. It doesn't necessarily mean we want to penalize a model. Rather, the evaluation should focus on the overall coherence, relevance, and completeness of the explanations in relation to the query and reasoning. The goal is to evaluate whether the model can generate explanations that are meaningful and contextually appropriate, not whether it can replicate specific elements from certain datasets.
> >     We hope our research can inspire more research in this area and contribute to the development of XAI, making it more interpretable, trustworthy and human-centered.
> >
> > **3. (W3) Clarification on Explanation:**
> > We understand your concern about the quality of our explanations. As we mentioned before, our reason-elements are derived from KGs and highlighted by the attention mechanism. It serves as a starting point for generating more refined and human-centered explanations. These reason-element can not represent the complete explanation, but are cues to guide the generation process of formulating the complete explanation. The main idea is that we utilize these reason-elements through the constrained generation instruction, we can "translate" the model's reasoning into natural, human-understandable language.
> >
> > As for the hallucination problem, it is a very common problem and a big topic in LLMs. We are not designing to remove all the hallucination, but using strong constrained instructions to narrow down the knowledge search space of LLMs, to ensure the trustworthy. Our instruction is to guide the LLMs to generate more relevant and focused explanations, thereby reducing the hallucinated content.
> >
> >
> > **4. (W4) Clarification on Contribution:**
> >
> > - **Current Focus and Contribution:** Our paper primarily focuses on enhancing the explainability of LLMs by integrating KGs and generating human-understandable explanations. Our claimed main contribution is transforming the "black-box" nature of LLMs into a more transparent and interpretable form, which aligns with human understanding.
> >
> > - **Application in RLHF (a Potential Extension):** We mention this in the paper as a potential extension of our work. We want to emphasize that this is not the main focus of our paper. We are not claiming that our current work focuses on solving the RLHF problem. Rather, we are proposing a potential extension of our work to address the RLHF problem. The mention is an example of how our work can be extended to contribute to the research community. We will modify the manuscript to clarify this point.
> >
> >
> > **Not yet finished, continue to the next comment...**

---

> > > ### Author Response · Authors · 2023-11-21
> > >
> > > **Response to Questions:**
> > >
> > > **1. (Q1) Method Effectiveness:**
> > > We will clarify the effectiveness of our method in the following aspects:
> > > - **Working Process:** The reasoning process of LLMs is complex and difficult to interpret. Our approach integrates KGs to narrow down the knowledge search space of LLMs. Through a more transparent and interpretable graph structure, we can better understand the decision-making process of LLMs. GAT is a typical method to learn the importance of nodes in a graph, and influence the reasoning process of LLMs. However, we acknowledge our limitations in the current design. The quality of KG (which is ConceptNet in our case) and more advanced GNN are important factors that can affect the effectiveness of our method. We believe that our approach and dataset are a step in the right direction towards making LLMs more interpretable and trustworthy.
> > > - **Human-centered Evaluation:** We included expert evaluation, human evaluation and automated evaluation to assess the quality of our explanations. Our evaluation includes eight dimensions, and is evaluated by 50 general users, 3 experts and gpt4/3.5. The results confirm the high quality and accuracy of our explanations. One expert commented that our explanations are "In comparison to prior explanations, these explanations provide a more intuitive understanding of the model's decision-making process. The explanations are cogent, and even in instances of erroneous predictions, the underlying reasoning remains transparent and comprehensible."
> > > - **Clarification on Accuracy:** Our final explanation is not used to affect the answer selection. The explanations are generated to reflect the decision-making process of the LLM in a human-understandable way. The accuracy of the explanations is also influenced by the LLM's performance. If the LLM's performance is low, the explanations will also be less accurate. However, it doesn't mean that our explanations are not accurate or not helpful. Our explanations can be used to debug the model.
> > >
> > > **2. (Q2) Emphasize Our Dataset:**
> > > Thank you for recognizing the value of our approach. We agree extending our work as a model for XAI is a promising direction. We appreciate the perspective you've offered. However, there are several key reasons why we emphasize the dataset in our paper:
> > > - **Filling a Crucial Gap in XAI:** Our primary motivation is to address a significant gap in XAI, specifically the lack of comprehensive datasets that can explain the decision-making process of LLMs in a human-understandable way. By focusing on the dataset, we aim to provide a valuable resource that not only demystifies the "black-box" nature of LLMs, but more importantly, can catalyze further research and development in XAI.
> > > - **Urgency and Importance of Our Dataset:** In the rapidly evolving field of AI, the urgency for robust and reliable datasets for XAI is paramount. Our dataset offers a unique and timely contribution that can immediately benefit researchers and practitioners in the field, helping them to develop more transparent, interpretable, and trustworthy AI systems.
> > >
> > > ---
> > > We are grateful for the opportunity to clarify our work. We hope that our response has addressed your concerns. We look forward to hearing from you.

---

> ### Author Response · Authors · 2023-11-22
> **Look Forward to Your Response**
>
> Dear Reviewer,
>
> As we approach the conclusion of the author-reviewer discussion period, we eagerly anticipate your thoughts on our rebuttal. We have endeavored to comprehensively address the concerns you raised. If there are any remaining issues or queries, we warmly welcome your input.
>
> We appreciate your feedback and look forward to your response.
>
> \
> Best regards,
>
> Authors

---

> > ### Comment · Reviewer_w11A · 2023-11-22
> > **Thank you for the response**
> >
> > I thank the authors for the detailed response.
> >
> > > "In the examples you mentioned, such as the inclusion of "play_mozart" for a question about music, it's important to recognize that models may sometimes establish connections that, while potentially relevant, might not align with every human interpretation"
> >
> > I think this is the core argument that I and multiple reviewers are trying to make. A dataset for "explainable AI", which would be used to evaluate how model explanations closely resembles human expectation/reasoning, should reflect human expectation/reasoning. We are not convinced that the current dataset has that quality. I fear that in the future, models which achieve super-human performance on this dataset can argue that they are explainable, while in reality that they are not.
> >
> > I urge the authors to do extensive human evaluation on the generated data and then release the dataset. For now, I am keeping my current decision.

---

> ### Author Response · Authors · 2023-11-22
>
> Dear Reviewer,
>
> Thank you for your reply. We understand and appreciate your concern regarding the alignment of our dataset with human reasoning and expectations. Your listed task is explanations for potential reasoning of how a correct answer is selected (irrelevant to models). This is not about how the model makes decisions, and is totally **different from our task**. Our task is to **explain the reasoning process of a decision model**.
>
> The "explainable AI" is a big topic, and our work here aligns with **explainable machine learning**, which is not to reflect human expectations, but to **interpret and explain machine learning models**. Many recent works have been done in this area, such as [1], [2], and [3].
>
> An example of [1] (dataset) from OpenAI, explaining why word "dollars" is generated:
>
> ```
> A broad effort is under way to understand what really works in health care, perhaps leading to better value for dollars spent.
>
> Explanation:
> Token: dollars
> mentions of "American" and related terms
> ```
> The weakness of [1] is that it can only explain a single word not complete reasoning, and it is still difficult for humans to understand the reasoning process of the LM.
>
> The key idea in this topic is to make the reasoning process of machine models more transparent, interpretable, and human-understandable. Our work is in this direction.
>
> We want to emphasize our novelty: (1) our dataset provides a new way of explaining how LLMs work (via more transparent surrogate components, e.g., reason-elements), and (2) explanations of LLMs' decision-making are more human-understandable compared to other works in this area.
>
> We have conducted a comprehensive human study to evaluate the quality of our explanations (Section 5). The results show that our explanations are human-understandable and can help users understand the reasoning process of the **decision model**.
>
> We hope this explanation addresses your concerns, and we are grateful for your feedback. Please let us know if you have any further questions or comments, we are happy to address them.
>
> \
> Sincerely,
>
> Authors
>
> ---
> [1] Steven Bills, Nick Cammarata, Dan Mossing, Henk Tillman, Leo Gao, Gabriel Goh, Ilya Sutskever, Jan Leike, Jeff Wu, and William Saunders. Language models can explain neurons in language models. https://openaipublic.blob.core.windows.net/neuron-explainer/ paper/index.html, 2023.
>
> [2] Weld, Daniel S., and Gagan Bansal. "The challenge of crafting intelligible intelligence." Communications of the ACM 62, no. 6 (2019): 70-79.
>
> [3] Lai, Vivian, Han Liu, and Chenhao Tan. "" Why is' Chicago'deceptive?" Towards Building Model-Driven Tutorials for Humans." In Proceedings of the 2020 CHI Conference on Human Factors in Computing Systems, pp. 1-13. 2020.

---

### Official Review · Reviewer_vkjW · 2023-11-09

**Soundness:** 3 good
**Presentation:** 2 fair
**Contribution:** 3 good
**Rating:** 5
**Confidence:** 2

**Summary:**

This work introduces a new explanation dataset for the Question Answering task by utilising knowledge graphs to build explanation components like why-choose and why-not-choose components, leveraging which will help in improving the performance of LLMs in this task. Graph attention and knowledge graphs are used to obtain the reason elements which are then converted to the why-choose and why-not-choose explanations which can be easily understood by humans. This dataset can be further used in RLHF which can improve the model’s performance. The dataset is constructed in the following manner :

1)The subgraph retrieved from the knowledge graph, g_k is pruned to obtain g_e which has only the important relations of the question and answer entities.

2)Node-type embeddings, feature embeddings of the previous layer and the relational embeddings are used to calculate the feature embeddings of the current layer. The final layer nodes mimic the decision-making process and are used as the reason elements.

3)Top nodes ranked by alpha(attention mechanism) are used as reason elements from the final layer of the graph and a generator model like GPT-3.5-turbo is used to generate the why and why-not explanations in a controlled manner.

The dataset used to build the explanation dataset is the CommonsenseQA dataset and the LLM used is RoBERTa-Large.

Extensive evaluation in understanding the overall quality, understandability, trustworthiness, satisfaction, sufficiency of detail, irrelevance, completeness and accuracy of the dataset is conducted using human evaluations. The authors also show how the performance of the model improves over the baseline, GPT-3.5 by using explanations in all three scenarios, vanilla, chain-of-thought and self-consistency methods.

**Strengths:**

This work introduces a novel explanation dataset which includes the why-choose and the why-not-choose explanations. These explanations can be better interpreted by humans and can improve the performance of the LLMs over the baseline method of not using the explanations in all types of scenarios, vanilla, chain-of-thought and self-consistency. This shows that these explanations are generalizable to most scenarios in improving the performance of the reasoning task.

**Weaknesses:**

As per my understanding, the why-not choose part requires the dataset to have multiple-choice question answers and is not generalisable to other scenarios(please correct me if I am wrong).

Minor comment : There is a mismatch in the explanation of the equation 8 and the actual equation 8, where m_ts and m_es is mixed up. More clarification on this is required and it would be better to include how the message is computed in the main text rather than the appendix

**Questions:**

1)What reasoning was used in choosing Roberta-Large as the LLM, M? Why was the masked language model used instead of autoregressive LLMs? Is it extendable to autoregressive models as well?

2)Can the intuition behind equation 4 in page 6 be explained clearly?

3)How can the claim that the LLM’s reasoning is very similar to “why” explanations in the dataset since it improves the performance be made? It could merely be providing additional context which helps the LLM to provide better answer

---

> ### Author Response · Authors · 2023-11-21
>
> Thank you for taking the time to review our paper and for the constructive feedback. We appreciate the opportunity to address your concerns:
>
>
>
>
> **Response to Weaknesses:**
>
> **1. (W1) Generalizability of Our Method:**
>
> We appreciate the opportunity to clarify the generalizability of the 'why-not choose' of our model.
>
> The 'why-not choose' component of our model is designed to provide explanations for not selecting certain options in a multiple-choice question setting. The intuition behind this design is to provide a more comprehensive explanation for the model's decision-making process. In a multiple-choice question (MCQ) setting, the model's decision is not only based on the correct answer but also on the incorrect options. Therefore, it is important to provide explanations for both the correct and incorrect options.
>
> While it is true that our current implementation focuses on MCQ scenarios, our approach can be adaptable to other formats. One of the key aspects of our methodology is the comparative analysis of different possible answers or outcomes. This can be applied to scenarios beyond MCQs, such as:
>
> - **Binary or Open-Ended Questions:** By slightly modifying the approach to compare the selected answer with potential alternatives or reasoning paths. For example, in a binary question, the model's decision can be compared with the alternative decision. In an open-ended question, the model's answer can be compared with the counterfactual answer.
> - **Decision-Making Scenarios:** Where the model can explain why certain options were not chosen based on a set of criteria or data points.
>
> We acknowledge that extending our 'why-not choose' approach to scenarios beyond MCQs is an interesting area for future research and development. We are actively exploring ways to adapt and apply this methodology to a broader range of question types and decision-making processes.
>
> It is important to highlight that the primary objective of our dataset is to explain the reasoning behind the LM's decision-making process. Our dataset is designed to enable
> general users to easily understand this process, which can be helpful for future human-centered applications. We believe that our paper makes a substantial contribution to the field by addressing a critical need for transparency and explainability in AI decision-making.
>
> **2. (W2) Clarification on Main Text:**
>
> We appreciate your attention to detail and agree that clarity in mathematical expressions is crucial for the understanding of our work. We apologize for the typos in our equations.
> We will address this issue as follows:
>
> - We will fix the typos in the revision.
> - Due to the page limit, we moved the detailed mathematical expressions to the appendix. We will revise the main text to provide a high-level overview of the mathematical expressions and refer the readers to the appendix for the detailed expressions.
> - To provide a clear understanding of the mathematical expressions, we will add more details of how the message is computed in the revised version. The message is computed by the node feature embedding, node embedding, and relation embedding, with a linear transformation. It is designed to capture the information of the node and its neighbors. The attention mechanism is used to aggregate the messages from neighbors. We will add the details in the revised version.
>
>
> **Not yet finished, continue to the next comment...**

---

> ### Author Response · Authors · 2023-11-21
>
> **Response to Questions:**
>
> **1. (Q1) Model Choice:**
>
> We selected RoBERTa-Large for several reasons:
>
> - RoBERTa is well-known for its robust performance across a variety of NLP tasks. Its widespread use in the NLP community makes it a reliable base model for evaluating new methods and approaches.
> - There are no human-understandable explanations for LLMs. RoBERTa is a fundamental model in the LLM family, we believe that it is important to start with such a model to establish a baseline for future research.
>
>
> Our current dataset primarily focuses on RoBERTa; however, it's important to emphasize that our methodology is universally applicable across a diverse range of LMs. The foundational elements of our approach, such as the integration of KGs and the interpreting component, are designed with flexibility in mind and are not specific to RoBERTa. These components can be adaptable to the architectures and functionalities of various LMs.
>
> Key aspects of our method, such as the extraction and utilization of knowledge from KGs, as well as the generation and validation of explanations, are inherently model-agnostic. This means they can be readily applied to other LLMs with minimal adjustments. As part of our future work, we plan to extend our dataset to include a variety of LMs.
>
> **2. (Q2) Intuition Behind Equation 4:**
>
> This equation is used to normalize scores obtained from our human study. In our human study, we aimed to engage participants effectively and obtain accurate evaluation results. To achieve this, we included a variety of questions, each measuring different dimensions of our dataset. Notably, one of these dimensions was 'irrelevance,' where a lower score indicated a better quality. This is opposite to the other dimensions, where a higher score indicates a better quality.
>
> We introduced Equation 4 to address this inconsistency and enhance the readability of our results. The primary goal of this normalization was to align all scores on a common scale where a higher value uniformly indicates better quality.
>
> **3. (Q3) Clarification on Reasoning:**
>
> Our model's reasoning is very similar to 'why' explanations in the dataset. This is because our model transforms the 'black-box' nature of LLMs into a more transparent graph structure, leading to reasoning in LLMs that is very similar to "why" explanations.
>
> - **Transformation to Transparent Structure:** The core of our approach lies in converting the opaque reasoning process of LLMs into a more transparent and structured form. This is achieved by integrating the retrieved graph into the LM's reasoning process.
> - **Narrowing Down Knowledge Search Space:** By leveraging KG, we effectively narrow down the knowledge search space for the LM. Our approach helps LM focus on the most relevant and contextually appropriate information, reducing irrelevant or inaccurate reasoning.
> - **Structured Knowledge for Better Interpretability:** The graph provides structured representations of knowledge, which are inherently more interpretable than unstructured data. We utilize this structured knowledge to help LM align its reasoning process more closely with the "why" explanations.
> - **Guiding the LM's Reasoning Process:** The graph not only provides structure but also guides the LM in its reasoning. The LM is constrained to utilize knowledge from the graph, which ensures that its reasoning is grounded in relevant knowledge space.
>
> ---
> We are grateful for the opportunity to clarify our work. We hope that our response has addressed your concerns. We look forward to hearing from you.

---

> > ### Comment · Reviewer_vkjW · 2023-11-22
> > **Q3 clarification**
> >
> > I believe the clarification for Q3 conveys that the reasoning generated by LLM is similar to the 'why' explanations in the dataset. However the LLM's 'reasoning' itself could be very different from the reasoning it generates, since the outputs for the question generated by the LLMs from these explanations do not necessarily correspond to the explanations. But the wording in the paper, makes it seem that the LLM's reasoning is very similar to the 'why' explanation, which is why I am confused

---

> ### Author Response · Authors · 2023-11-22
> **Look Forward to Your Response**
>
> Dear Reviewer,
>
> As we approach the conclusion of the author-reviewer discussion period, we eagerly anticipate your thoughts on our rebuttal. We have endeavored to comprehensively address the concerns you raised. If there are any remaining issues or queries, we warmly welcome your input.
>
> We appreciate your feedback and look forward to your response.
>
> \
> Best regards,
>
> Authors

---

> ### Author Response · Authors · 2023-11-22
>
> Dear Reviewer,
>
> Thank you for your reply. We appreciate the opportunity to clarify this further.
>
> In our paper, we have a **decision model**, and a **generator model**. Our task is to explain the decision-making process of the **decision model**. I am assuming that you are referring to the **generator model** when you say "the LLM". The generator model is used to generate explanations based on the key reason-elements extracted from the decision model. The generator model is not used to make decisions, it is like a **"translator"** that translates the decision model's reasoning process into human-understandable explanations. More specifically, the "why explanation" is reflected how the decision model reasons, it should be "very similar" to the real **decision model's** reasoning, not the **"generator model"**.
>
> Under our strong constraint, the generator model will NOT reason the question and answer or make decisions. We will add this clarification in the revised version.
>
> Our work aims to make the reasoning process of LLMs **(decision model)** more transparent and interpretable, acknowledging that the explanations are approximations designed to be as faithful as possible to the LLM's **(decision model)** outputs.
>
> We hope this explanation addresses your concerns, and we are grateful for your feedback. Please let us know if you have any further questions or comments, we are happy to address them.
>
> \
> Sincerely,
>
> Authors

---

### Meta-Review · Area_Chair_osEk · 2023-12-06

**Metareview:**

This paper introduces a new explanation dataset for question answering. It contains 12k question-answer-explanation triples, generated by 1) extracting reason-element using a graph-attention network  model and a pruned KG, and 2) generating explanations given the reason elements and the correct answer with GPT-3.5. The quality of the dataset is verified by human and automatic eval metrics (with GPT-3.5 and GPT-4). It also shows that incorporating explanations significantly improves reasoning evaluation.

While the reviewers appreciate the novelty of the approach used to construct the dataset, almost all reviewers have raised concerns around the quality and the generalizability of the dataset. In particular, reviewers w11A and ZCRc questioned the decision of using RoBERTA-large for generating the reasoning elements, and were concerned that it might bias evaluation results. Reviewer MEtr also pointed out the the intermediate reason-elements were not validated, and might have quality issues.

**Justification For Why Not Higher Score:**

Almost all reviewers have raised concerns about potential quality issue of the dataset, and the choice of using RoBERTA-large for extracting the intermediate reasoning elements.

**Justification For Why Not Lower Score:**

N/A

---

### Decision · Program_Chairs · 2024-01-16

Reject